# GRK2-Dependent HuR Phosphorylation Regulates HIF1α Activation under Hypoxia or Adrenergic Stress

**DOI:** 10.3390/cancers12051216

**Published:** 2020-05-13

**Authors:** Clara Reglero, Vanesa Lafarga, Verónica Rivas, Ángela Albitre, Paula Ramos, Susana R. Berciano, Olga Tapia, María L. Martínez-Chantar, Federico Mayor Jr, Petronila Penela

**Affiliations:** 1Department of Molecular Biology and Molecular Biology Centre Severo Ochoa (CMBSO), the Spanish National Research Council, the Autonomous University of Madrid (UAM-CSIC), 28049 Madrid, Spain; cr3023@cumc.columbia.edu (C.R.); vlafarga@cnio.es (V.L.); vrivas@cbm.csic.es (V.R.); angela.albitre@cbm.csic.es (A.A.); pramos@cbm.csic.es (P.R.); srberciano@cbm.csic.es (S.R.B.); fmayor@cbm.csic.es (F.M.J.); 2Institute for Cancer Genetics, Columbia University Medical Center, New York, NY 10032, USA; 3Department of Molecular Oncology, Spanish National Cancer Research Centre (CNIO), 28029 Madrid, Spain; 4Department of Cellular and Molecular Mechanisms in Inflammatory and Autoimmune Diseases, Institute of Health Research La Princesa, 28006 Madrid, Spain; 5Department of Anatomy and Cell Biology, CIBER of Neurodegenerative Diseases (CIBERNED), University of Cantabria–IDIVAL, 39011 Santander, Spain; olga.tapia@unican.es; 6CIC bioGUNE, Center for Cooperative Research in Biosciences, Liver Disease and Liver Metabolism Lab, 48160 Derio, Spain; mlmartinez@cicbiogune.es; 7Biomedical Research Center Network of Hepatic and Digestive Diseases (CIBERehd), The Instituto de Salud Carlos III (ISCIII), 28029 Madrid, Spain; 8CIBER of Cardiovascular Diseases (CIBERCV), The Instituto de Salud Carlos III (ISCIII), 28029 Madrid, Spain

**Keywords:** hypoxia, β-adrenergic signaling, breast cancer, mRNA regulation, nucleocytoplasmic shuttling, GRK2, HuR, HIF1α, VEGF

## Abstract

Adaptation to hypoxia is a common feature in solid tumors orchestrated by oxygen-dependent and independent upregulation of the hypoxia-inducible factor-1α (HIF-1α). We unveiled that G protein-coupled receptor kinase (GRK2), known to be overexpressed in certain tumors, fosters this hypoxic pathway via phosphorylation of the mRNA-binding protein HuR, a central HIF-1α modulator. GRK2-mediated HuR phosphorylation increases the total levels and cytoplasmic shuttling of HuR in response to hypoxia, and GRK2-phosphodefective HuR mutants show defective cytosolic accumulation and lower binding to HIF-1α mRNA in hypoxic Hela cells. Interestingly, enhanced GRK2 and HuR expression correlate in luminal breast cancer patients. GRK2 also promotes the HuR/HIF-1α axis and VEGF-C accumulation in normoxic MCF7 breast luminal cancer cells and is required for the induction of HuR/HIF1-α in response to adrenergic stress. Our results point to a relevant role of the GRK2/HuR/HIF-1α module in the adaptation of malignant cells to tumor microenvironment-related stresses.

## 1. Introduction

Oxygen is a key factor in embryonic development and in normal tissue homeostasis, and its shortage triggers adaptive mechanisms required for cell survival [1,2]. Hypoxia is also a hallmark of solid tumors that results from a leaky vasculature within the tumor and the high proliferation rate of both cancerous cells and immune infiltrating cells [3]. Tumor-associated hypoxia correlates with cellular resistance to chemotherapy, metastasis, and poor patient survival, indicating that hypoxia acts as an extrinsic driver of tumor progression [4,5]. The hypoxia-inducible transcription factors HIF-1 and HIF-2 are the main coordinators of the cellular response to low oxygen tension in both normal and transformed cells [2,5]. HIF-1 is formed by two helix-loop-helix-PAS proteins, namely HIF-1α and HIF-1β or ARNT. Under normoxia, HIF-1α subunits are efficiently ubiquitinated and targeted for proteasome-dependent degradation by the multimeric SCF2 E3 ligase coupled with the von Hippel-Lindau tumor suppressor protein VHL, which recognizes HIF subunits only after their hidroxylation by PHD1-3, a family of oxygen-dependent prolyl-hidroxylases. In hypoxic conditions, HIF-1α subunits are stabilized as a result of PHD inhibition and translocated to the nucleus, where they dimerize with ARNT, in order to activate the transcription of hundreds of genes [5,6].

Interestingly, in cancerous cells, the HIF signaling pathway can also be regulated by other environmental and intracellular signaling factors in an oxygen-independent manner, via modulation of protein stability, gene transcription, or mRNA binding proteins such as YB-1 and HuR (Human antigen R), which are often up-regulated in tumor contexts thereby leading to enhanced translation of HIF-1α mRNA in normoxic and hypoxic conditions [7,8,9]. HIF-1α is often downstream of the activation of different plasma membrane receptors-triggered signaling pathways, thus linking HIF-1α status to the growth needs of cells [10,11,12].

Different G protein-coupled receptors (GPCRs) have been associated with the progression of several types of tumors [13,14]. Of note, activation of GPCRs by β-adrenergic, endothelin-1 or lysophosphatidic acid agonists has been shown to promote the stabilization of HIF proteins [12,15,16], thus mimicking hypoxic conditions. Cumulative evidence indicates that chronic neurosympathetic activity fosters the malignant growth and dissemination of tumor cells through direct stimulation of β2AR receptors [17]. Several processes appear to underlie the influence of βAR on tumor progression, including increased HIF-1α signaling and angiogenesis. Besides its role in GPCR desensitization, the G protein-coupled receptor kinase 2 (GRK2) is emerging as a key signaling hub. GRK2 can act as a positive effector of certain GPCR and receptor-tyrosine kinases (RTK) transduction cascades [18,19], and also directly interacts/phosphorylates components of signaling networks involved in cell transformation [20]. We have recently shown that GRK2 expression is up-regulated in different breast cancer contexts, playing a driving role in the acquisition of oncogenic features [21,22]. Concurrently, GRK2 downregulation takes place in the breast tumor vessels [23]. Such opposite and cellular-specific alterations in GRK2 expression seem to be causally related, promoting a marked increase of intra-tumoral hypoxia and of the tumor-associated macrophage-derived factor adrenomedullin, a known VHL-HIF-1 target gene.

In this report we unveil that GRK2 is a relevant modulator of the hypoxic pathway in transformed epithelial cells via phosphorylation-dependent modulation of the mRNA-binding protein HuR, leading to subsequent nuclear accumulation of the HIF-1α protein. Stimulation of GRK2 upon hypoxia or adrenergic receptor activation emerges as a novel mechanism of regulation of HIF signaling in physiological or pathological processes, such as cancer progression.

## 2. Results

Our previous results pointed to GRK2 as a common regulatory and transducing effector of diverse pathways that are altered in luminal and basal breast cancer [21,22]. Remarkably, the mRNA binding protein ELAV-like protein 1 (HuR), a key upstream modulator of HIF-1α and other targets, is also overexpressed in these tumor contexts [24,25]. Moreover, GRK2 upregulation alters the activity of several transcription factors in normal breast cells that are chronically stimulated with EGF [21], whereas HuR activity changes in response to EGF [26]. A subset of EGF- and GRK2-regulated transcription factors in such conditions can be linked to a GRK2-mediated gain-of-function of the prolyl isomerase Pin1 [21] (Appendix A). Interestingly, other subsets can be linked to HuR (Appendix A), since their mRNAs are either regulated or potentially targeted by this protein [27]. Altogether, these observations suggested a potential functional relationship between HuR and GRK2 in both tumor and non-tumor settings.

### 2.1. HuR Protein Levels Positively Correlate with GRK2 Activity

We analyzed the steady-state levels of HuR protein under different GRK2 expression and activation status contexts in Hela cells, a cellular model widely used in the functional characterization of HuR. The protein levels of HuR directly correlated with GRK2 abundance, being significantly upregulated in cells over-expressing the wild-type GRK2 (Hela-WT5), but down-modulated upon stable GRK2 silencing (Hela-shGRK2) (Figure 1A). Interestingly, HuR abundance remained unaltered compared to parental Hela cells, upon stable over-expression of the catalytically inactive GRK2-K220R mutant (Hela-K1 cells) or of GRK2-S670A, which lacks the ability to phosphorylate a subset of GRK2 substrates (Hela-A1 cells) [21,28]. These results suggested that GRK2-mediated phosphorylation of either HuR or an intermediate factor was involved in the regulation of HuR expression levels.

### 2.2. HuR Is a GRK2 Phosphorylation Substrate

Purified GST-HuR was efficiently phosphorylated by recombinant GRK2 (Km of ~ 48 nM, Figure 1B), similar to the well-known physiological substrates of GRK2 [28,29], whereas no phosphorylation was observed in the recombinant GRK2-K220R (Figure 1C), indicating that HuR is a direct target of GRK2. Consistently, a direct and preferential binding of HuR to the catalytic domain of GRK2 was detected in the overlay assays (Appendix A).

Akin to some GRK2 substrates such as HDAC6 [28], phosphorylation of GRK2 at the regulatory site Ser670 seems to be required to enable kinase activity towards HuR, since the recombinant GRK2-S670A mutant was incapable of phosphorylating HuR (Figure 1C), despite being able to fully phosphorylate other well-established GRK2 substrates [28]. These data and those obtained in Hela-A1 cells suggest that HuR belongs to the subset of ‘phospho-Ser670-biased’ GRK2 targets.

We identified three potential phosphorylation site(s) on the GRK2-phosphorylated GST-HuR, by using proteomic approaches (Appendix A). Single or double site-directed mutagenesis to alanine of these candidate sites, followed by in vitro phosphorylation assays showed that GST-HuR-Thr-142/143A and GST-HuRS-197A purified proteins displayed a significantly reduced phosphorylation by GRK2, compared to the wild-type (Figure 1D), indicating that these residues are accounting for circa 75% of total GRK2-dependent HuR phosphorylation. Interestingly, these residues are located in two key functional and regulatory regions of the HuR protein, the second RNA-binding domain (RRM2) (residues Thr142 and 143) and the nucleocytoplasmic localization hinge region (residue Ser197) (Appendix A). Phosphorylation of the hinge region and the RRM domains by different kinases has been shown to underlie changes in HuR subcellular localization, binding affinity with mRNA, and regulation of translational efficiency [30,31,32,33,34].

### 2.3. GRK2 Activity Modulates the Hypoxia-Induced Modulation of HuR Cellular Levels and Cytoplasmic Shuttling

Hypoxia is a well-characterized stress known to upregulate HuR protein levels, in order to foster HuR actions [35]. Interestingly, Hela-WT5 cells stably expressing GRK2, displayed an enhanced boost in HuR levels upon acute exposure to low oxygen, while such HuR upregulation was absent upon kinase downregulation (Hela-shGRK2 cells) (Figure 2A). A similar unresponsive pattern was observed in the hypoxic Hela cells expressing GRK2 mutants that are unable to phosphorylate HuR (Hela-A1 and Hela-K1 cells) (Figure 2A). These data supported the notion that regulation of HuR by GRK2 was strictly dependent on its kinase activity and on previous GRK2 phosphorylation at Ser670. Consistently, parallel to changes in the HuR levels, a clear up-regulation of S670-GRK2 phosphorylation was noted after 2 h of hypoxia, which was sustained afterwards in both parental and Hela-WT5 cells (Figure 2B), but not in Hela-A1 or Hela-K1 cells.

HuR is mainly localized in the cell nucleus in basal conditions, whereas its activity in stress conditions is tightly linked to protein cytosolic shuttling [32,36]. Akin to stress conditions such as UV treatment [37], serum starvation [38], or heat shock [39], cytosolic translocation of the HuR protein occurs after prolonged CoCl_2_ exposure, a condition mimicking hypoxic stress [35]. We observed that GRK2-mediated HuR phosphorylation markedly affects the HuR cytoplasmic accumulation in hypoxia. In parental Hela cells, HuR protein is mobilized from the nucleus as early as 2 h after low oxygen exposure, in line with previous reports [40], and enhanced cytoplasmic HuR levels are still noted 24 h afterwards (Figure 3A). Such a pattern of HuR mobilization was potentiated by over-expression of wild-type GRK2, whereas no shuttling was observed in the cells expressing a catalytically inactive GRK2 mutant (Hela-K1) or upon GRK2 downmodulation (Hela-shGRK2). Consistent with a relevant role for GRK2-mediated HuR phosphorylation in this process, the cytosolic redistribution of phospho-defective HuR mutants HuR-S197A or HuR-T142/143A was markedly reduced upon exposure to hypoxia for 3 h or 6 h (Figure 3B). The combination of these mutations in the triple HuR-T142/143/197A construct resulted in a stronger impairment of HuR shuttling (Figure 3B). None of these HuR mutations appear to disturb the localization of HuR in untreated cells, suggesting that GRK2 regulation specifically takes place under stress.

Overall, these results indicated that GRK2-mediated phosphorylation at the residues located in the different domains of HuR can modulate the cellular levels and the cytoplasmic localization of HuR protein, in response to hypoxia.

### 2.4. GRK2 Phosphorylation of HuR Is Required for HIF-1α Upregulation in Response to Hypoxia

Central to cellular hypoxia is the upregulation of HIF-1α, which is in turn required for the transcriptional induction of genes involved in metabolism, angiogenesis, or survival [5,6,41]. In addition to protein stabilization, HuR-mediated translational control of HIF-1α mRNA is an important mechanism accounting for HIF-1α protein accumulation, in response to short-term hypoxia [42,43]. Interestingly, nuclear levels of HIF-1α after 4 h of hypoxia were markedly potentiated in the presence of extra GRK2 (HeLa-WT5 cells), compared to parental Hela cells. By contrast, such HIF-1α nuclear accumulation was impaired in the Hela cells overexpressing either GRK2-K220R or GRK2-S670A mutants, both unable to phosphorylate/mobilize the HuR protein (Figure 4A).

These results suggested that GRK2-phosphorylated HuR could be more efficient in binding to HIF-1α mRNA or in regulating its decay or translation rates.

Then, we performed RNA immunoprecipitation (RIP) assays to detect HuR-HIF-1α mRNA complexes in Hela cells that were overexpressing HA-tagged constructs of HuR. As shown in Figure 4B, circa 2.7-fold enrichment in HuR-bound HIF-1α mRNA was found in the hypoxic cells, as compared to the normoxic cells. In contrast, no significant hypoxia-induced increase in the binding of the HuR-T142/T143A-S197A mutant to HIF-1α mRNA was noted. The specificity of these responses was assessed using the non-HuR target GAPDH mRNA and control IgG immunoprecipitates, in parallel experiments (Figure 4B). These data strongly supported the importance of GRK2-mediated HuR modulation in the activation of the HuR/HIF-1α axis in hypoxia.

It has been proposed that stress-induced HuR might not bind with identical affinities to its mRNA targets [40], suggesting an HuR target “prioritization” in response to specific stress stimuli. Interestingly, the hypoxia-induced protein levels of p21, a well-known HuR target [34] was not altered by the HuR-T142/T143A-S197A expression (Figure 4C), thereby suggesting that phosphorylation of HuR by GRK2 alone or together with other hypoxia-induced kinases, does not necessarily affect other HuR targets.

### 2.5. GRK2 Fosters the HuR/HIF-1α Axis and a Pro-Lymphangiogenic Response in the Normoxic Breast Luminal Cancer Cells

Expression of the *ELAVL1* (HuR) and *ADRBK1* (GRK2) genes showed a direct correlation in breast cancer patients inspected with the web tool CANCERTOOL (Appendix A). A significant positive association in hormone-stratified cohorts was found only in estrogen-positive tumors (Appendix A). No association of *ELAVL1* and *ADRBK1* co-expression was evidenced with the molecular subtypes of breast cancer “basal-like”, “normal-like” or “HER2-enriched” in the three independent cohorts, whereas a significant direct correlation was noted in the luminal subtype (Appendix A).

Therefore, we used the cell line MCF7, a model of estrogen receptor (ER)-positive and luminal A human breast tumors engineered to express different dosages of GRK2 [21], and further characterized the effects on HuR. In basal conditions, cells overexpressing GRK2 (MCF7-GRK2) displayed a marked increase in total HuR protein levels, compared to parental MCF7 cells, whereas the opposite effect was observed upon GRK2 down-regulation in MCF7-shGRK2 cells (Figure 5A). Basal cytosolic translocation of HuR was higher in MCF7-GRK2 cells compared to parental cells, as inferred from the significant reduction of nuclear HuR levels, in spite of the global upregulation of the protein (Figure 5B).

This effect depended on the GRK2-dependent phosphorylation of HuR, as the inducible over-expression of GRK2-S670A did not trigger a reduction in nuclear HuR levels, whereas a kinase mutant mimicking the positive regulatory S670 phosphorylation event (GRK2-S670D) decreased the nuclear HuR levels akin to the wild-type GRK2 protein (Figure 5C).

We next investigated whether GRK2-dependent HuR modulation in MCF7 cells might affect HIF-1α nuclear distribution. As expected in tumor cells [44], normoxic MCF7 cells had a constitutive elevation of HIF-1α protein in the nucleus. Levels of HIF-1α protein display cell-to-cell variations, with a Gaussian distribution when the cells were stratified in subgroups, according to staining intensity, ranging from no signal (0) to maximal signal (800), with steps of 100 arbitrary units, peaking at the average intensity of 364.40 ± 39 (mean ± SD) arbitrary units (mean gray value). We then grouped the stratified cell population according to the proportion of cells with intensities in the range of the average value (34.8 ± 2.4%), above the average value (11.7 ± 4.8%), or below the average value (53.5 ± 6.1%) (Figure 5D). Interestingly, the overexpression of GRK2 did not affect the average intensity of nuclear HIF-1α protein (382.58 ± 17 arbitrary units), but markedly altered the distribution of cells around this value. Thus, the proportion of cells with HIF-1α levels higher than the average value was increased circa 3-fold, while those with lower levels were reduced circa 5.5-fold. Conversely, in the MCF7-shGRK2 cells, the average intensity of the HIF-1α protein (422.25 ± 23) was within the range of the parental cells, but the proportion of cells with HIF-1α a level over this value was reduced 2-fold (Figure 5D). Altogether, these results suggested that the increased expression of GRK2 noted in the breast cancer cell lines and in patients, would also contribute to tumor progression by fostering the cellular mechanisms that induce and stabilize the HIF-1α protein in normoxia.

A main pro-carcinogenic effect of the HIF-1α factor is the induction of angiogenesis and lymphangiogenesis through the activation of transcription of the VEGF family members [41,45]. Of note, the secretion of VEGF-C, a potent inducer of lymphatic vessels through which breast cancer cells often disseminate [46,47] was strongly increased in the MCF7 cells overexpressing GRK2 as compared to the parental cells, and conversely secretion was reduced upon GRK2 downregulation (Figure 5E). Therefore, it is tempting to suggest that GRK2 might impact VEGF-C expression via pathways downstream of HuR phosphorylation to foster lymphangiogenesis and metastases of luminal breast cancer cells.

### 2.6. Adrenergic Stress Induces a HuR/HIF1-α Hypoxia-Like Response in MCF7 Breast Cancer Cells in a GRK2-Mediated Manner

The effects of GRK2 on the HuR/HIF1-α axis in MCF7 cells suggested that stressful or microenvironment conditions other than hypoxia might also contribute to foster this tumorigenic pathway. In this context, it is known that adrenergic stimulation promotes ERK-mediated phosphorylation of GRK2 on Ser670 in different cellular systems [48] and also strengths hypoxic signaling via HIF1-α accumulation in a GRK2-dependent manner in normoxic endothelial cells [12]. In addition, compelling evidence demonstrates that chronic adrenergic stress is involved in breast cancer progression [49,50].

Therefore, we addressed whether chronic β2-adrenergic receptor stimulation promoted HuR regulation by GRK2 in MCF7 breast cancer cells. Isoproterenol challenge upregulated total HuR protein levels in both parental and MCF7-GRK2 cells, but not in GRK2-silenced cells (Figure 6A). Isoproterenol also triggered parallel changes in GRK2 protein, suggesting that adrenergic effects on HuR levels involve GRK2. In addition, isoproterenol stimulation also promoted a reduction of HuR nuclear protein levels in parental cells, and this effect was potentiated in the presence of extra kinase in MCF7-GRK2 cells (Figure 6B). However, isoproterenol treatment failed to promote the nuclear exit of HuR in MCF7-shGRK2 cells, indicating that GRK2 activity is required for HuR mobilization. Consistent with the increased mobilization of nuclear HuR, nuclear HIF1-α protein was upregulated after 48 h of βAR stimulation in cells overexpressing GRK2, compared to parental cells (Figure 6C).

Overall, our results point that the GRK2/HuR/HIF-1α axis is potentiated in response to hypoxic and adrenergic stresses, thus emerging as a relevant pro-tumorigenic module.

## 3. Discussion

In this paper, we identified GRK2 as a modulator of the HuR/HIF-1α axis in relevant pathological settings. We uncovered that HuR is a GRK2 substrate, requiring prior phosphorylation of GRK2 at its Ser670 regulatory site. Phosphorylation by GRK2 enhances the HuR levels and the cytosolic shuttling of HuR upon hypoxic conditions, favoring its binding to HIF-1α mRNA and increased nuclear accumulation of this central modulator of hypoxia-driven pathways. In the MCF7 cells, a cellular model of luminal A breast cancer in which upregulation of GRK2 and HuR frequently concur, GRK2 increased the total and basal cytosolic HuR protein levels, which correlated with a higher nuclear distribution of HIF-1α and enhanced secretion of the lymphoangiogenic factor VEGF-C in normoxic conditions. Moreover, chronic adrenergic stimulation fosters the HuR/HIF1-α module in MCF7 cells through GRK2 (Figure 7), which could facilitate survival of malignant cells even before the expanding tumor mass become hypoxic, and could also improve the adaptation of tumor cells to the hypoxic environment of lymphatic vessels and to other tumor-related stresses.

Ser670 phosphorylation is an emerging regulatory switcher in GRK2 substrate specificity and partner association [18]. Context-specific ERK1/2-dependent GRK2 Ser670 phosphorylation is necessary for GRK2 to phosphorylate HDAC6 [21,28], to disrupt its interaction with GIT1 [51], or to allow its HSP90-mediated localization to the mitochondrial membrane [52]. A variety of stimuli, including growth factors [28,29], adrenergic and other GPCR ligands [53], ischemia [54], or hypoxic conditions (this manuscript) can trigger GRK2 phosphorylation at this residue via ERK1/2 stimulation, which is also triggered by hypoxic mimetics [55]. Therefore, it is likely that enhanced ERK1/2 activity in response to different environmental factors would switch on the ability of GRK2 to phosphorylate and modulate HuR.

We have identified residues within the hinge region (S197) and the RRM2 domain (T142/T143) as the main targets of GRK2-mediated HuR phosphorylation, and mutations preventing such modifications specifically impair hypoxia-induced HuR cytoplasmic shuttling in HeLa cells, without altering its basal localization. Moreover, the triple HuR-T142/143/197A mutant also displays reduced binding to HIF-1α mRNA in hypoxic conditions.

Several mechanisms might underlie the observed effects of GRK2 on HuR functionality. Enhanced cytosolic levels of GRK2-phosphorylated HuR might be a consequence of increased nuclear export/decreased nuclear import or fostered cytoplasmic retention. The long hinge region (aa 187–243) includes the HuR Nucleocytoplasmic Shuttling (HNS) sequence (aa 205–237) containing both nuclear localization and nuclear export determinants [30,31,32,33,56]. This HNS resembles a consensus bipartite Nuclear Localization Signal, NLS sequence, with two clusters of the basic residues near the N-terminus (aa 205–206 and aa217–219), being the first of the ones critical for nuclear targeting [57]. Introduction of a negative charge by phosphorylation at S197 might reduce the electrostatic potential of the neighboring basic lysine residues (aa 205–206), which are key for the interaction with the import machinery and favors the nuclear exit. Additionally, phosphorylation of HuR on S197 might inhibit its interaction with trans-acting factors that facilitate its translocation through the nuclear pore complex [30,56], thus favoring cytoplasmic retention. The hinge region of HuR (aa 190–243) is involved in the interaction with the nuclear protein ligands pp32 and April [58], in turn recognized by the export receptor CRM1 [39], thereby making it feasible for the phosphorylation of S197 to improve its association to pp32/April. Our results showing a cytoplasmic localization of HuR phosphorylated on S197 adds complexity to these HuR trafficking regulatory mechanisms.

GRK2-mediated phosphorylation at T142/143 within the RRM2, the second HuR RNA binding domain also fosters HuR cytoplasmic localization. It has been reported that interaction of HuR with nuclear importin TNR2 is reduced by binding of ARE-containing RNA to HuR [56]. Since GRK2 phosphorylation increases the formation of HuR/HIF1 mRNA complexes, this could hamper the interaction with the nuclear import machinery. In addition to TNR2 and the transporting homologue TNR1 [56,59], the importing α/β pathway is required for nuclear import of HuR [60]. We cannot rule out that the mechanisms whereby GRK2 regulates HuR shuttling include interference of HuR/importin α/β complexes.

The subcellular location in which the phosphorylation of HuR by GRK2 takes place could shed light on the GRK2-mediated modulation of HuR trafficking. HuR and GRK2 display opposite canonical subcellular distribution at the basal conditions. Although GRK2 was initially described as a cytoplasmic protein, its nuclear presence has also been reported [61], thus allowing HuR phosphorylation at this location and regulation of nuclear export. Alternatively, or in addition, since HuR moves to the cytosol under hypoxia, this would facilitate GRK2-mediated phosphorylation and decreased nuclear import or fostered cytoplasmic retention.

Preventing HuR modification by GRK2 also results in a defective interaction with HIF1-α mRNA. HuR phosphorylation has a notable impact on the binding of HuR to mRNA targets [24]. Conformational changes occurring on RRM1 and RRM2 regions are crucial for mRNA-binding [62,63]. Tandem RRM1-RRM2 displays prominent inter-domain flexibility, particularly affecting residues in the linker and in the loops of RRM [63]. This flexibility allows HuR to fit around low affinity strands with high efficiency, enabling the recognition of many different mRNA substrates. Interestingly, Thr142 is one of the positions in the RRM2 loop showing large chemical shift perturbations [63]. Thus, phosphorylation of Thr142 by GRK2 could favor structural flexibility in the close-bound state of HuR by preventing some stabilizing interaction in the proximity of the inter-domain linker. It has been proposed that binding of mRNAs bearing shorter 3′UTR with lower AU content could be more influenced by the flexibility of RRM1-2 domains [63]. Remarkably, with 8 AU-rich motifs throughout the 3′UTR (AREsite2 database, http://rna.tbi.univie.ac.at/AREsite), HIF1α mRNA could fit in the group of mRNAs with greater dependence on HuR conformations. Therefore, it is tempting to suggest that GRK2 might favor recognition of a subset of mRNAs displaying lower intrinsic affinities for HuR.

Furthermore, although we have not analyzed in-depth the mechanism whereby HuR phosphorylation by GRK2 impacts its total protein levels, it is worth noting that phosphorylation of RRM2 residues has been linked to HuR stabilization. Phosphorylation of Thr118 by CHK2 protects HuR from proteasomal degradation upon heat shock [64]. Modifications at this site alter the extent of HuR polyubiquitination, mainly assembled on Lys182, a RRM2 residue key for protein stability. Whether phosphorylation of Thr142/143 by GRK2 has similar effects remains to be investigated.

The role of HuR in breast cancer development is well-documented in model cell lines, in vivo animal models, and clinical studies [24,25]. The increased levels of HuR in MCF7 cells compared to non-tumoral cells is implicated in the regulation of many genes involved in the cell cycle, survival, and angiogenesis [38,65,66]. Our data put forward that in this luminal A breast cancer model GRK2 would foster HuR/HIF-1α pathways even in normoxic conditions. In MCF7 cells, GRK2 dosage positively correlated with the total HuR protein levels, cytoplasmic HuR translocation, higher nuclear distribution of HIF-1α, and increased VEGF-C secretion. Upregulation of HIF-1α in normoxia is a feature of tumor cells [44] and different non-canonical modalities of HIF-1α modulation have been identified [8]. We have previously shown that in MCF7 cells the activation of estrogen and growth factor pathways converge for promoting an enhanced GRK2 expression and fostering GRK2 phosphorylation at Ser670 [21], which would enable GRK2-meditaed HuR phosphorylation and enhanced functionality and, thus, cooperate with other pathways in upregulating HIF-1α cascades.

HuR can modulate HIF1α mRNA via the 3′UTR region for transcript stabilization or via 5′UTR for translation, depending on the cellular context [67,68]. It seems that the 5′UTR-based mechanism predominates in highly malignant cells, while those based on 3′UTR-mediated stabilization do so in less aggressive ones (such as MCF7) [69]. Thus, it is feasible that GRK2 phosphorylation of HuR in MCF7 cells would foster the latter mechanism. Our results also point that the GRK2/HuR/ HIF1α axis is involved in modulating the expression of VEGF-C, a critical factor for remodeling the lymphatic network and tumor cell dissemination [45,46,47]. It is well-known that VEGF factors are transcriptional targets of HIF-1α but are also HIF-1-independent post-transcriptional targets of HuR itself [70,71]. Therefore, GRK2-phosphorylated HuR might also directly modulate the VEGF-C mRNA in breast tumor MCF7 cells.

Adrenergic stress is emerging as a relevant microenvironmental player in cancer progression [49,50]. Chronic emotional stressors increase the catecholamine levels and βAR activation in cancer cells and increases metastatic potential [72] and levels of pro-angiogenic factors [73]. We unravel that chronic adrenergic stimulation triggers the upregulation of both GRK2 and HuR protein levels, increases cytoplasmic HuR and fosters nuclear HIF-1α presence, thus, mimicking a pseudo-hypoxia situation that might contribute to tumor progression. Chronic adrenergic stimulation increases GRK2 expression by different transcriptional and post-transcriptional mechanisms [74,75], and also to promote HuR expression [76]. Normoxic HIF-1α accumulation in adrenergic-stimulated endothelial cells depends on the phosphorylation of β-AR by GRK2 [12]. Our data suggest a mechanism linking β-AR activation to a GRK2/HuR/HIF1α downstream axis (Figure 7). Upon agonist stimulation, GRK2-mediated phosphorylation of β-AR would enable β-arrestin-dependent MAPK activation and subsequent modification of GRK2 at S670, the key event allowing the targeting of HuR and the HIF1 cascade. β-adrenergic stimulation is known to activate the MAPK cascade [77,78], and prolonged and high-dose stimulation of the β2AR induces a switch in receptor coupling from Gs to Gi, shifting the signaling to MAPK pathways [79].

In sum, our data put forward a relevant role for the GRK2/HuR/HIF1α module in cancer cells and in response to the adrenergic overdrive, regardless of the oxygen status. Enhanced GRK2 expression and S670 phosphorylation status in the given tumor contexts would lead to enhanced HuR functionality, thus, counteracting the canonical HIF protein degradation driven by prolyl hydroxylases [7,9] and recapitulating a pseudo-hypoxic state in normoxia, as a consequence. Besides inducing the angiogenic remodeling of tumoral stroma, GRK2-induced pseudo-hypoxia might favor cancer cell de-differentiation and emergence of cancer stem-like cells. Pseudohypoxic cells are often observed at the tumor–stromal interface in locally invading tumors, providing growing advantages during tumor expansion, as these cells are more aggressive and resistant [4]. The coordinated regulation of key players in angiogenesis such as HuR, HIF-1α, and VEGF-C suggest that upregulation of GRK2 might be a relevant event for acquiring increased malignancy and invasiveness, and point that combined pharmacological inhibition of GRK2 might be useful in certain cancer therapy contexts. This mechanism reinforces the notion of GRK2 as a central oncomodulator influencing several hallmarks of cancer [22].

## 4. Methods

### 4.1. Cell Culture and Treatments

Cells were maintained in DMEM supplemented with 10% fetal bovine serum. Stable Hela (ATCC) cells were previously described [27]. MCF7luc-F5 cells stably expressing pcDNA3-GRK2-wt or pLKO-GRK2-shRNA (5′-GCAAGAAAGCCAAGAACAAGC-3′) and tetracycline-inducible expression (TET-on) system for mutant GRK2-S670A or GRK2-S670D in MCF7 cell, were also previously described [21]. GRK2 expression in the TET-on system was promoted by treating the cells with 1 µg/mL tetracyclin for 24 h. Hypoxia (1% O_2_) was achieved in an Invivo2 400 hypoxia Workstation (Baker Ruskinn, USA). Isoproterenol (50 μM) was supplemented with 1 mM ascorbic acid and 20 mM HEPES pH 7.5.

### 4.2. Plasmids and Cell Transfection

Plasmids encoding HA-HuR WT were previously described [33]. HA-HuR-S197A, HA-HuR-T142/143A, or HA-HuR-T142/143A-S197A were generated using QuickChange^®^ Lightning site-directed mutagenesis kit (Thermo Fisher, Waltham, MA, USA). HeLa cells (70%–80% confluent monolayers in 60 or 100 mm dishes) were transiently transfected using the lipofectamine/plus method.

### 4.3. Immunoprecipitation, Western Blot, Dot Blot, and ELISA

Whole cellular lysates were prepared in RIPA buffer (Tris-HCl 20 mM pH 7.5, 150 mM NaCl, 1% Triton-X100, 0.1% SDS, 0.5% sodium deoxycholate, cocktail of protease, and phosphatase inhibitors), as described [28]. For the analysis of nuclear and cytoplasmic pools of HuR and HIF-1α, cells were collected in hypotonic lysis buffer (10 mM Tris–HCl, pH 7.4, 10 mM NaCl, 3 mM MgCl2, 0.3% (v/v) Nonidet P-40, 2 mM Na3V04, 10 mM NaF, and protease inhibitors), incubated on ice for 10 min and centrifuged at 500 ×g for 5 min to obtain the nuclear fraction. The cytoplasmic supernatant contains the extranuclear fraction (plasma membrane, microsomal vesicles, cytoskeleton, and cytosol). Pellets containing cell nuclei were washed in lysis buffer without Nonidet P-40 and again pelleted at 500 ×g. Both fractions were solubilized in Laemmli sample buffer and resolved by SDS–PAGE. Proteins were resolved in SDS–PAGE and transferred to nitrocellulose membranes. Cytosolic (GAPDH) and nuclear (Nucleolin) markers were used for control loading. For the dot-blot experiments, GST-tagged GRK2 fragments (2 µg) and recombinant GRK2 full-length protein were spotted on strips of nitrocellulose and incubated for 4 h at 37 °C, with GST-HuR (2 µg/mL) in the interaction buffer (20 mM Tris-HCl pH 7.5, 100 mM NaCl, 1 mM EDTA, 0.3% NP-40, 10% Glycerol, 0.5 mM NaF, and 1 mM Na_3_V0_4_). Membranes were incubated with the indicated primary antibodies—α-Actin (1:2000, Santa Cruz Biotechnology, Santa Cruz, CA, USA), GAPDH (1:2000, Santa Cruz Biotechnology), (pSer670-GRK2 (1:500, GeneTex, Irvine, CA, USA), GRK2 (1:1000, Santa Cruz Biotechnology), HuR (1:000, Santa Cruz Biotechnology), Nucleolin (1:1000, Santa Cruz Biotechnology), p21 (1:500, Santa Cruz Biotechnology), and α-Tubulin (1:2000, Santa Cruz Biotechnology). Blots were developed using a chemiluminescent method (ECL, Amersham, Little Chalfont, UK). Band density was quantitated by laser densitometric analysis.

VEGF-C secretion was determined in cell-conditioned media of MCF7 cells incubated in DMEM 1% FCS for 48 h, by using a Human VEGF-C Quantikine ELISA Kit (R&D Systems, Minneapolis, MN, USA). Measurements in duplicates were normalized to the amount of the total protein in cell lysates quantified by the Lowry method.

### 4.4. Immunofluorescence and Confocal Microscopy

As described in [28], 4% PFA-fixed cells were processed and incubated O/N at 4 °C, with primary antibodies—HA (1:600, Cell Signaling, Leiden, The Netherlands), HIF1α (1:200, Novus Biologicals, Centennial, CO, USA) and HuR (1:200, Santa Cruz Biotechnology). Then, coverslips were incubated for 1 h RT with fluorescent secondary antibodies, and for 5 min with To-Pro3 (1:2000, ThermoFisher, Waltham, MA, USA) or DAPI (1 µg/mL, Merck, Darmstadt, Germany) to stain nuclei. Finally, samples were mounted using Mowiol-DABCO (Boston Bioproducts, Ashland, MA, USA), let to dry overnight at RT and then stored at 4 °C. Images were acquired using a confocal laser microscope LSM710 (Zeiss, Oberkochen, Germany) and the nucleus area and mean intensity values were analyzed using ImageJ (NIH, Bethesda, MD, USA). In both the basal and stress conditions, nuclear masks were drawn using the nuclear staining distribution pattern of HuR and were guided by the morphological features (nucleolus, euchromatin, speckles) (Appendix A). For each cell, three measurements were made within the nucleoplasm, excluding the HuR-negative areas corresponding to nucleoli and nuclear speckles, or indistinctly in the cytoplasm peripheral to the nucleus. Fluorescence intensity values were corrected for background staining. The ratio of the mean cytoplasmic (C) to nuclear (N) value was calculated for each cell. Average C/N values from all cells within a field were pooled for subsequent plotting and comparative analysis. Cells from 10 different randomly acquired fields (40× or 63×) were measured for each cell type and condition. Average values from all cells within a field were pooled for subsequent plotting and analysis.

### 4.5. Kinase Activity Assays

In vitro kinase assays with wild-type, S670A or K220R GRK2 full-length (50 nM); and WT, T142/143A or S197A GST-tagged HuR recombinant proteins (50–250 nM for Km value and 100 nM for the rest) were performed, as described [28]. The Michaelis constant (Km) was estimated by double-reciprocal plot analysis of three independent experiments.

### 4.6. Mass Spectrometry Analysis

GRK2 (50 nM) and GST-HuR (100 nM) were incubated in the presence of cold ATP, as detailed above and resolved by SDS–PAGE. The band corresponding to full-length GST-HuR was digested by trypsin and chymotrypsin, and the fragments were analyzed using LC–MS/MS in the ‘CBMSO PROTEIN CHEMISTRY FACILITY’ that belonged to the ProteoRed network, PRB2-ISCIII, supported by the grant PT13/0001.

### 4.7. RNA Immunoprecipitation

RNA Immunoprecipitation (RIP) assays were performed, as previously described [80]. Whole-cell lysates (500 µg) were incubated with a suspension of Protein A/G PLUS-Agarose, precoated with 15 µg of either IgG or anti-HA antibody-conjugated agarose beads (all from Santa Cruz Biotechnology). Bound mRNA was purified from immunoprecipitants, retrotranscribed by RT-PCR, and the HIFα mRNA content was measured by real-time PCR analysis and normalized to GAPDH mRNA bound in a nonspecific manner to IgG.

### 4.8. Bioinformatics Analysis of HuR and GRK2 Expression in Human Breast Tumors

Pairwise correlation of gene expression *ELAVL1* (HuR protein) and *ADRBK1* (GRK2 protein) levels in breast tumors was calculated and represented using the CANCERTOOL webtool [81]. Pearson correlation test was applied to analyze the relationship between paired genes, and the correlation coefficient and *p*-values were adjusted using the Benjamini–Hochberg method. Datasets with a correlation coefficient greater than 20% (−0.2 < R < 0.2) and a *p*-value lower than 0.05 were considered.

### 4.9. Statistical Analysis

Data analysis was performed using the GraphPad Prism for Windows. Means between groups were compared by two-way or one-way ANOVA with Bonferroni’s or Tukey’s post-hoc test, or with unpaired Student’s *t*-test, as indicated in the figure legends. All results are expressed as mean ± SEM.

## 5. Conclusions

An active GRK2/HuR/ HIF-1α module, particularly in estrogen-positive luminal A breast tumors could facilitate survival of malignant cells, even before the expanding tumor mass becomes hypoxic, and also improves the adaptation of tumor cells to the hypoxic environment of lymphatic vessels and to other tumor-related stresses, such as chronic over-activation of the adrenergic system, in order to foster tumor progression.

## Figures and Tables

**Figure 1 cancers-12-01216-f001:**
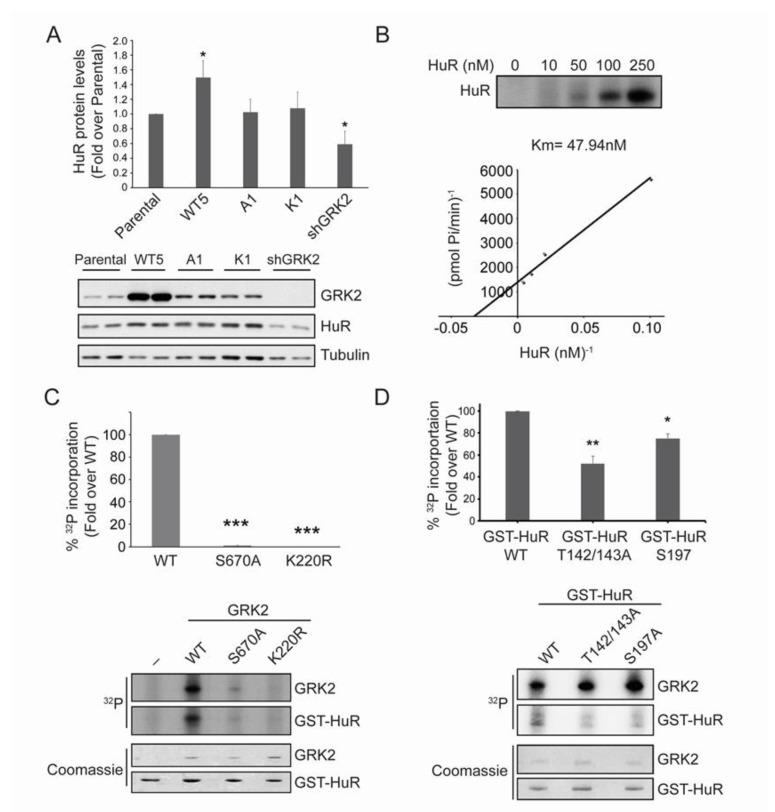
GRK2 phosphorylates and regulates HuR. (**A**) HuR total levels were analyzed by immunoblotting of the total lysates of parental and HeLa cell lines over-expressing wild-type GRK2 (Hela-WT5), a mutant lacking the ability to phosphorylate a subset of GRK2 substrates (Hela-A1), the catalytically inactive GRK2-K220R mutant (Hela-K1) or harboring stable GRK2 silencing (Hela-shGRK2). Values are mean ± SEM from four independent experiments. * *p* < 0.05 vs. parental (Student’s *t*-test). (**B**) GRK2 and GST-HuR were incubated in the presence of [γ-^32^P]-ATP, as detailed in Materials and Methods. Km was estimated by double-reciprocal plot analysis of three independent experiments. (**C**) Phosphorylation of GST-HuR was performed in the presence of [γ-^32^P]-ATP using recombinant GRK2-WT, GRK2-S670A, or GRK2-K220R proteins, as described in Materials and Methods. Intensity of ^32^P and the Coomassie bands were quantified by densitometry and plotted as a percentage of WT GRK2-triggered ^32^P incorporation. Data representative of two independent experiments are shown. *** *p* < 0.001 vs. parental (Student’s *t*-test). (**D**) GRK2 (50 nM) and GST-HuR, GST-HuR-T142/143A, or GST-HuR-S197A (100 nM) were incubated in the presence of [γ-^32^P]-ATP, as detailed in Materials and Methods. Intensity of the ^32^P and Coomassie bands was quantified by densitometry and plotted as a percentage of GRK2-triggered ^32^P incorporation. The graph shows fold differences in two independent experiments. * *p* < 0.05 vs. WT ** *p* < 0.01 vs. WT (1-way ANOVA). Detailed information about the Western blots can be found in Appendix A.

**Figure 2 cancers-12-01216-f002:**
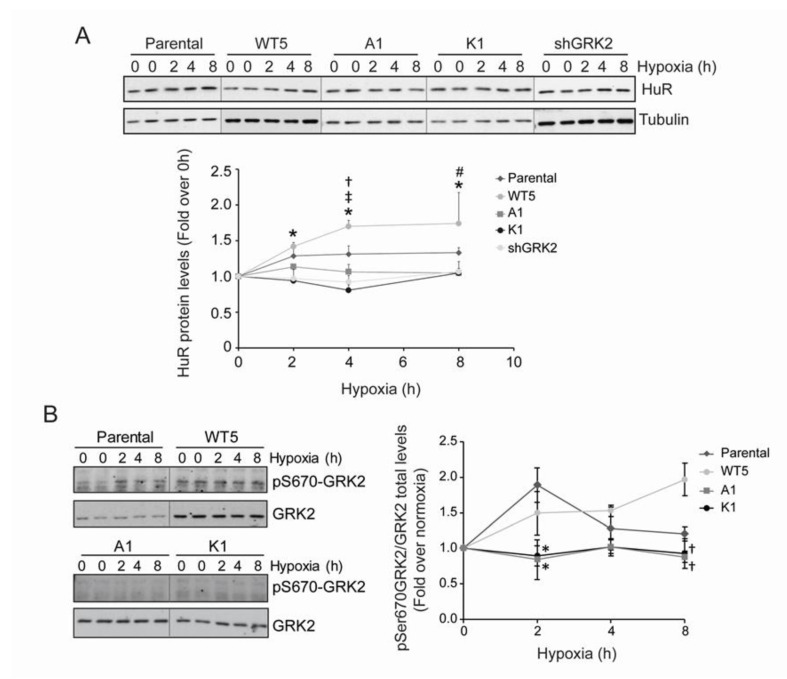
pS670-GRK2 modulates hypoxia-induced HuR upregulation. (**A**,**B**) HeLa stable cell lines were cultured under hypoxia and the pSer670 GRK2, GRK2, and HuR levels were analyzed by immunoblotting, at the indicated times. Values are mean ± SEM from 4–6 independent experiments. Upper panel: ‡ *p* < 0.05 wt5 vs. parental; # *p* < 0.05 A1 vs. parental; * *p* < 0.05 K1 vs. parental; † *p* < 0.05 shGRK2 vs. parental (Student’s *t*-test). Lower panel: * *p* < 0.05 A1 and K1 vs. parental; † *p* < 0.05 A1 and K1 vs. WT5 (Student’s *t*-test). Detailed information about the Western blots can be found in Appendix A.

**Figure 3 cancers-12-01216-f003:**
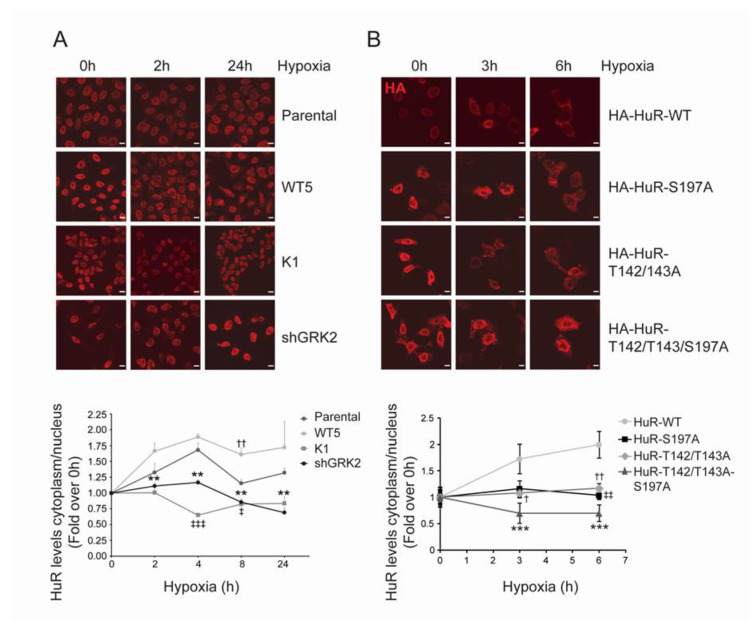
GRK2 affects hypoxia-dependent changes in HuR subcellular distribution. (**A**) The indicated HeLa stable cell lines were cultured under hypoxia, and HuR nuclear and cytosolic levels were analyzed by immunofluorescence, at the indicated times. Values are mean ± SEM from 3–4 independent experiments. †† *p* < 0.01 wt5 vs. parental; ‡ *p* < 0.05 ‡‡‡ *p* < 0.001 K1 vs. parental; ** *p* < 0.01 shGRK2 vs. parental (Student’s *t*-test). Representative images are shown. Scale Bar = 10 µm. (**B**) HeLa cells were transiently transfected with HA-HuR WT, HA-HuR-S197A, HA-HuR-T142/143A, or HA-HuR-T142/143A-S197A, and was cultured under hypoxia for the indicated times. Localization of HA-tagged HuR was assessed by immunofluorescence and the nuclear and cytosolic levels were quantified. Values are mean ± SD from two independent experiments. ‡‡ *p* < 0.01 S197A vs. wt; † *p* < 0.05 †† *p* < 0.01 T142/142A vs. wt; *** *p* < 0.001 T142/143A-S197A vs. wt; (two-way ANOVA). Representative images are shown. Scale Bar = 7 µm.

**Figure 4 cancers-12-01216-f004:**
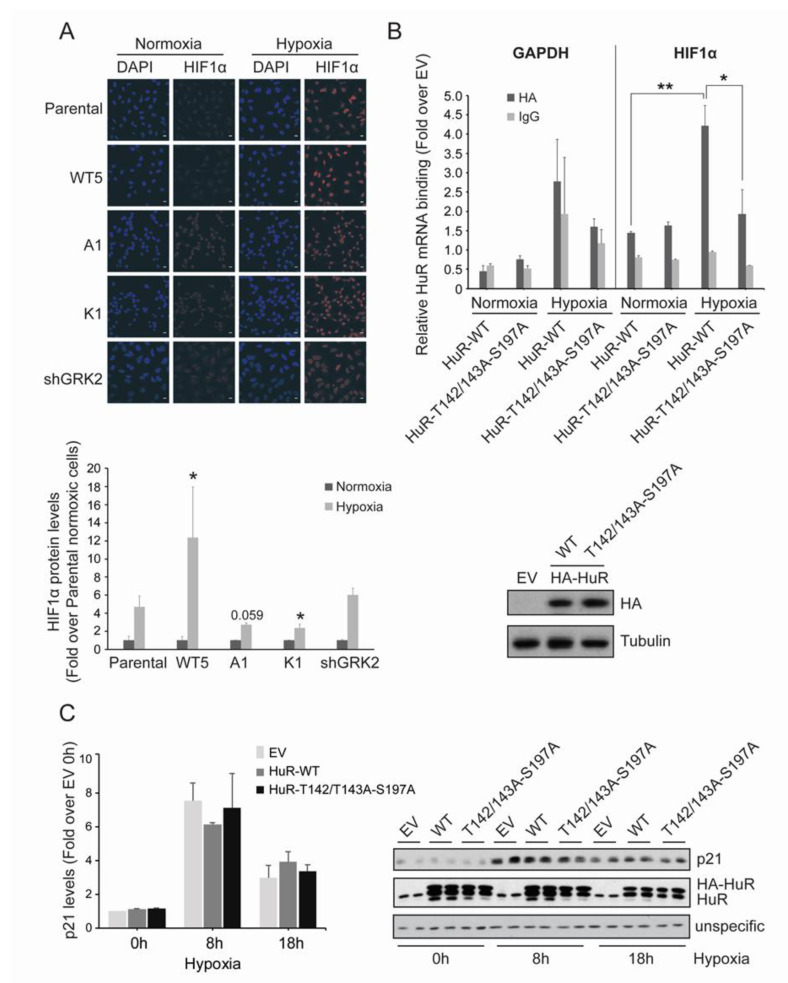
GRK2 kinase activity is required for HuR-induced upregulation of HIF1α. (**A**) HeLa stable cell lines were cultured under hypoxia for 2 h and HIF1α levels were analyzed by immunofluorescence. Values are mean ± SEM (fold-change over parental cells in normoxia) from 3–4 independent experiments. * *p* < 0.05 (Student’s *t*-test) comparing the changes in hypoxic HIF1α levels in Hela stable cells lines, with those observed in hypoxic parental cells. Representative images are shown. Scale Bar = 10 µm. (**B**) HeLa cells were transiently transfected with pcDNA3 as empty vector (EV), HA-HuR WT, or HA-HuR-T142/143A-S197A, and cultured for 4 h under hypoxia. Total lysates were then immunoprecipitated with either an HA antibody or control IgGs. RNA was purified from immunoprecipitates and used for qRT-PCRs. GAPDH was used as an endogenous control. The graph shows fold differences in transcript abundance (mean ± SEM in two independent experiments performed in triplicates). ** *p* < 0.01 hypoxic vs. normoxic WT; * *p* < 0.05 hypoxic triple mutant vs. hypoxic WT (two-way ANOVA). The expression levels of the different HA-HuR proteins analyzed by immunoblotting are shown below. (**C**) HeLa cells were transiently transfected with pcDNA3 as empty vector (EV), HA-HuR WT, or HA-HuR-T142/143A-S197A, and was cultured under hypoxia for the indicated times. p21 and endogenous and overexpressed HuR levels were analyzed by immunoblot. A non-specific band was used as the loading control. Values are mean ± SEM from two independent experiments. Detailed information about the Western blots can be found in Appendix A.

**Figure 5 cancers-12-01216-f005:**
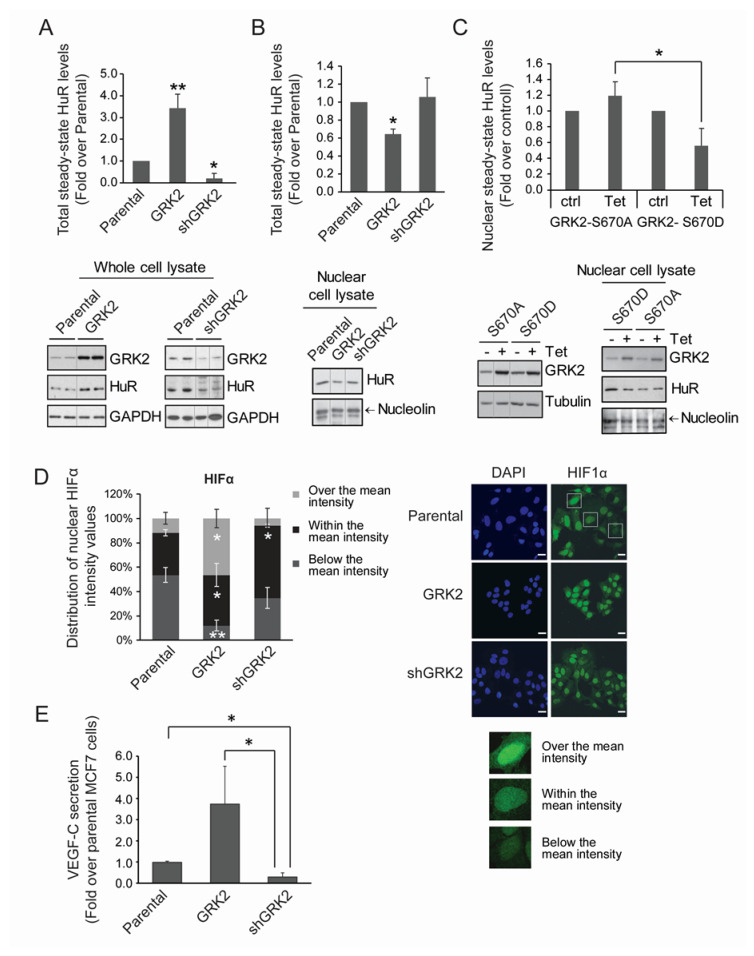
GRK2 modulates HuR levels and cytoplasmic shuttling, HIF1-α distribution, and VEGF-C secretion in luminal breast cancer cells. (**A**,**B**) Whole lysates or nuclear extracts from MCF7 cells stably expressing GRK2-wt or shGRK2 were used to analyze the HuR levels by immunoblot. Values are mean ± SEM from 3–4 independent experiments. * *p* < 0.05 ** *p* < 0.01 (Student’s *t*-test). Representative blots with loading controls are shown. (**C**) MCF7 stable inducible cell lines were treated with tetracyclin (Tet) to induce the expression of the indicated GRK2 mutants. HuR nuclear levels were analyzed by immunoblot. Values are mean ± SEM from two independent experiments. * *p* < 0.05 (Student’s *t*-test). Representative blots with loading controls are shown. (**D**,**E**) HIF1-α distribution and VEGF-C secretion levels were analyzed in parental MCF7 cells or in derived cells lines stably overexpressing GRK2 or a silencing construct (shGRK2) of the kinase. (**D**) HIF1-α and DAPI staining was analyzed by immunofluorescence; distribution of intensity values is shown. Values are distribution percentages of cells with HIF1-α intensity values over, within, or below the mean intensity. Representative images are shown. * *p* < 0.05; ** *p* < 0.01 (Student’s *t*-test) when each category (over, within, below) in MCF7-GRK2 and MCF7-shGRK2 cells is compared to parental ones. Scale Bar = 25 µm. (**E**) ELISA-measured VEGF-C secretion values are shown (mean ± SEM from 3 independent experiments). * *p* < 0.05 (Student’s *t*-test). Detailed information about the Western blots can be found in Appendix A.

**Figure 6 cancers-12-01216-f006:**
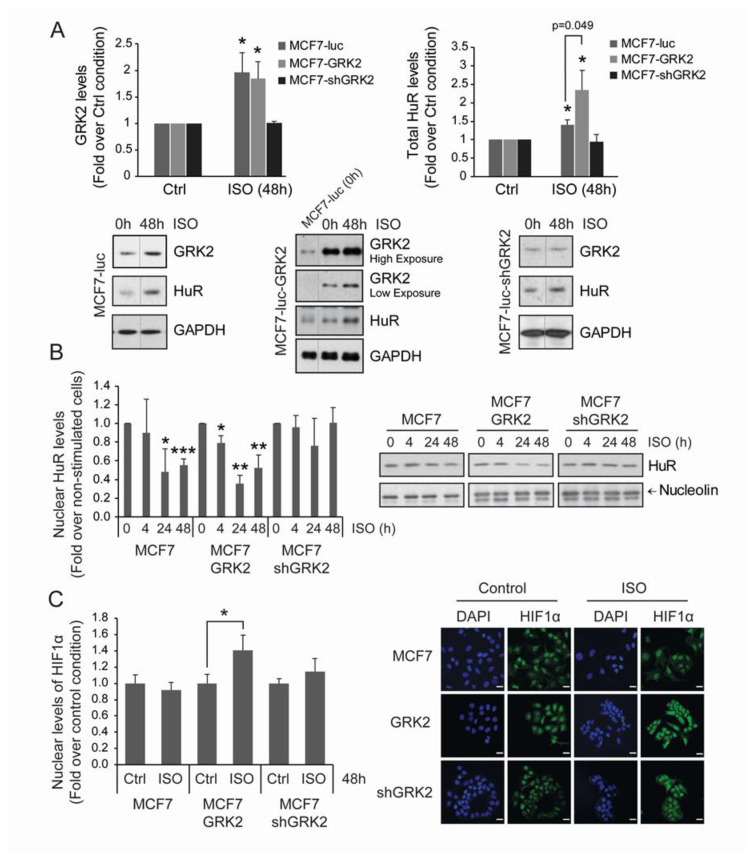
Chronic adrenergic stress leads to HIF1α accumulation through GRK2 and HuR upregulation in luminal breast cancer cells. Adrenergic signaling facilitates nuclear export of HuR in a GRK2-dependent manner. MCF7-luc (control MCF7 cells) and cells stably expressing GRK2-wt or shGRK2 were treated with 50 μM isoproterenol for the indicated times to induce adrenergic stress. Total GRK2 and HuR levels (**A**) and nuclear HuR levels (**B**) were analyzed by immunoblot with the indicated loading controls. (**C**) HIF1-α and DAPI nuclear marker levels were analyzed by immunofluorescence. Representative images are shown (scale bar, 25 μm). All values are mean ± SEM from three independent experiments. * *p* < 0.05, ** *p* < 0.01, ****p* < 0.001 (Student’s *t*-test). Detailed information about the Western blot can be found in Appendix A.

**Figure 7 cancers-12-01216-f007:**
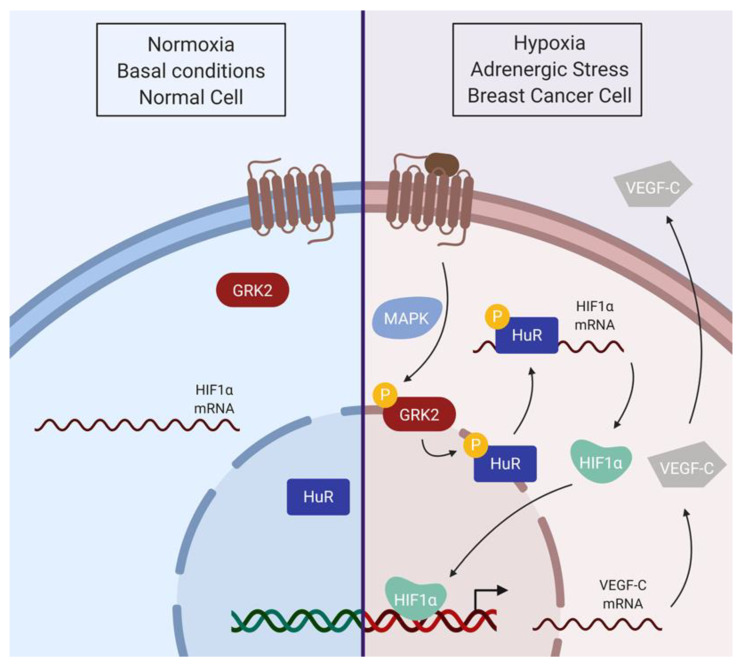
Model for the GRK2-dependent activation of the HuR/HIF1α axis. In basal conditions, the main HuR protein pool is located in the nucleus. Hypoxia or adrenergic signaling facilitates GRK2 phosphorylation at Ser670, enabling HuR phosphorylation by GRK2, thus increasing HuR nuclear export. Cytosolic HuR protein binds to and stabilizes HIF1-α mRNA, increasing protein translation and HIF1-α function as a transcription factor. As a consequence, transcription and protein secretion of the VEGF-C angiogenic factor would be enhanced (scheme created with BioRender).

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
