# Peer review of "GRK2-Dependent HuR Phosphorylation Regulates HIF1α Activation under Hypoxia or Adrenergic Stress"

_cancers, 2020, doi:10.3390/cancers12051216_

Round 1

Reviewer 1 Report

Reglero et al. revealed the GRK2/HuR/HIF-1alpha pathway in cancer cell lines. The authors' data is interesting. however, some points should be addressed before publication.

<Major comments>

  1. In figure 3, the authors show HuR staining (A) and HA staining (B). The authors describe the localization of these molecules based on figure 3. However, the authors did not show nuclear staining. Therefore, the description is not reliable. The authors must show nuclear staining, such as DAPI staining, in combination with the immunostaining.

  1. In the graph of figure 4A, the detail of statistical analysis is unclear. What the authors compare to? vs Normoxia? vs Parental?

  1. In Figure 5E, the author should perform statistical analysis between parental and shGRK2, because in line 285, the authors describe "conversely reduced upon GRK2down-regulation".

  1. In figure 6A, why the authors did not perform Actin staining as an internal control in MCF7-luc-GRK2 cells? If the author normalized data by different internal controls in figure 6A (Actin for MCF7-luc and MCF7-luc-shGRK2, and GAPDH for MCF7-luc-GRK2), the graphs showed in figure 6A is not scientific.

  1. In figure 7, VEGF should be changed to VEGF-C.

  1. In line 413-414, what does "(revised in [8])" mean?

<Minor comment>

  1. In figure 3A and 4A, the size of nuclei of A1 and K1 cells looks smaller than that of the parental cells. Do the authors know the reason?

  1. In line 418, is "Hur" "HuR"?

Author Response

Reviewer 1

Reglero et al. revealed the GRK2/HuR/HIF-1alpha pathway in cancer cell lines. The authors' data is interesting. however, some points should be addressed before publication.

We appreciate the comments of the reviewer, which have contributed to clarify and improve the manuscript.

Major comments

  • In figure 3, the authors show HuR staining (A) and HA staining (B). The authors describe the localization of these molecules based on figure 3. However, the authors did not show nuclear staining. Therefore, the description is not reliable. The authors must show nuclear staining, such as DAPI staining, in combination with the immunostaining.

We agree with the reviewer that for distinguishing subcellular compartments when a given protein is distributed among cytoplasmic and nuclear sites with complex distribution patterns, localization markers are usually needed. In fact, ther immunofluorescences corresponding to Figure 3 were performed by simultaneously staining of Hur protein (with HuR or HA anti-antibodies) and nuclei with DAPI or Topro-3, since we routinely include a nuclear marker for making easier cell localization under microscopy. However, for quantification of HuR in particular, DAPI images were not photographed in parallel because the pattern of HuR signal was clear enough by default to mark out the area of nuclei as detailed below. Given the current restrictions derived from the Covid19 pandemic situation to gain access to research labs and microscopy equipment in our center, we are unable now to take these images in our preparations (even if DAPI intensity was preserved over time) or to carry out additional immunofluorescence experiments for a representative image. 

However, we are confident that the HuR signal, measured as cytoplasmic versus nuclear in both basal and hypoxia conditions, reflects faithfully these subcellular compartments. The steady-state localization of HuR in Hela cells in basal conditions is clearly nuclear, displaying a well-established and strong signal in the nucleoplasm and being absent in the nucleoli. Although in transformed cell lines robust HuR expression at basal levels is often accompanied by protein accumulation in the cytoplasm, HuR remains predominantly in the nucleus of unstressed Hela cells. Therefore, staining of HuR itself is a reasonable marker for nucleus identification in Hela cells in basal conditions. Stress relocates a portion of HuR to the cytoplasm with a granulated pattern. However, even upon stress a significant amount of HuR remains in the nucleus (easily detected by varying laser strength), making it feasible to exploit such signal as a tool for nuclear identification (Kodiha M, Stochaj U (2012) Nuclear Transport: A Switch for the Oxidative Stress - Signaling Circuit? Journal of Signal Transduction, 2012:208650; Su H, Kodiha M, Lee S, Stochaj U (2013) Identification of Novel Markers That Demarcate the Nucleolus during Severe Stress and Chemotherapeutic Treatment. PLoS ONE 8(11): e80237). For consideration of the reviewer, the procedure of quantification is detailed below with representative images of the Figure 3. If the reviewer judges it of interest, this information about the quantification procedure can be added to the manuscript in the Material and Methods section.

  • In the graph of figure 4A, the detail of statistical analysis is unclear. What the authors compare to? vs Normoxia? vs Parental?

Statistical analysis in figure 4A involves a comparison of each one HIF1α data (expressed as fold change over Parental cells in Normoxia) in the different hypoxic-treated stable cell lines with the fold change detected in Parental cells in such hypoxic conditions. We have made this point clearer in the figure legend of Figure 4A lanes 206-207

  • In Figure 5E, the author should perform statistical analysis between parental and shGRK2, because in line 285, the authors describe "conversely reduced upon GRK2down-regulation".

As indicated by the reviewer, we have performed the required statistical analysis, which is now included in new figure 5E. We used the values of normalized VEGF-C (pg per mg of cellular protein) corresponding to parental (control) cells in the different experiments, so variability of the control was included when taken such values as reference (1-fold) for the other cellular conditions in each experiment.  VEGF-C levels are significantly different between parental and sh-GRK2 MCF7 cells (p= 0.0257, two-tailed T-test, unpaired), and between GRK2-overexpressing and shGRK2 cells, whereas only a trend is noted between parental and GRK2 overexpressing MCF7 cells (p=0.0745, two-tailed T-test, unpaired). This is due to the high range of fold change between experiments in GRK2-overexpressing MCF7 cells, despite in all cases the amount of VEGF-C secreted was clearly higher compared to parental cells.  

  • In figure 6A, why the authors did not perform Actin staining as an internal control in MCF7-luc-GRK2 cells? If the author normalized data by different internal controls in figure 6A (Actin for MCF7-luc and MCF7-luc-shGRK2, and GAPDH for MCF7-luc-GRK2), the graphs showed in figure 6A is not scientific.

We thank the reviewer for noting the inclusion of such different loading controls (actin and GAPDH blots in the same panel) as it may be confusing.  Routinely, nitrocellulose membranes of these experiments were immunoblotted to detect GRK2 and two markers for protein loading (usually Actin or Tubulin and GAPDH) in a sequential way and after stripping if necessary, to monitor the comparable overall expression of proteins in the stable cell lines. In the former 6A panel, Actin blots were selected as control for GRK2 blots for the picture of MCF7-luc and MCF7-luc-shGRK2 cells because in these particular experiments they were developed at the time of GRK2 blots in the autoradiography, while GAPDH data were subsequently obtained after stripping from the Actin blot (as shown in the corresponding uncropped blots in the Supplementary material section). However, in most cases GAPDH blots are probed before Actin and together with GRK2. However, irrespective of the order in GAPDH or actin probing, the data used for internal normalization were those from GAPDH. Following the suggestion of the reviewer, we have now changed of the loading control protein marker in the representative blots, so they are standardized to GAPDH in figure 6A, and also in figure 5A (following the suggestion of Reviewer 3 in the same sense). As indicated above, this change does not involve modifications in data quantification or plotted data inasmuch as densitometric values of GRK2 were always normalized by GAPDH in both panels 5A and 6A.

  • In figure 7, VEGF should be changed to VEGF-C

The term VEGF-C has been included in the figure 7 as indicated by the reviewer.

  • In line 413-414, what does "(revised in [8])" mean?

The words “revised in” were added to emphasize that reference [8] is a review contribution covering in depth non-canonical modalities of HIF-1α modulation (Iommarini, L.; Porcelli, A.M.; Gasparre, G.; Kurelac, I. Non-Canonical Mechanisms Regulating Hypoxia-Inducible Factor 1 Alpha in Cancer. Front Oncol 2017, 7, 286). However, for the sake of simplicity we have opted to delete “revised in” from now line 424 

Minor comments

  • In figure 3A and 4A, the size of nuclei of A1 and K1 cells looks smaller than that of the parental cells. Do the authors know the reason?

We agree with the reviewer that such a size difference is apparent, and we think this is an interesting observation. We have noted smaller nuclei in different stable clones and pooled Hela cells overexpressing GRK2-S670A or GRK2-K220R mutants but not of other GRK2 mutants, suggesting that is not an artefact of the selection processes but a specific effect event linked to the scaffold functions of catalytically inactive GRK2.  Unfortunately, we do not know yet what such functions may be. We have quantified the nuclear area and found that nuclei of Hela A1 and K1 are 74.2 µm2± 16.2 (n= 41) and 74.6 µm2 ± 30.5 (n=54) versus parental Hela cells 166µm2± 40.1 (n= 54). Interestingly, we have previously reported that the whole cell area of Hela A1 and K1 was also lower. These cell lines displayed a defective cell spreading compared to Hela cells caused by hyper-acetylation of microtubules, as GRK2 (but not the A or K mutants) phosphorylates and activates the tubulin deacetylase HDAC6 (Lafarga V, Aymerich I, Tapia O, Mayor F Jr, Penela P. A novel GRK2/HDAC6 interaction modulates cell spreading and motility. EMBO J. 2012 Feb 15;31(4):856-69). Thus, cellular area was 548.58µm2 ±66.47 in Hela versus 380.93µm2 ± 31.96 and 337.71 µm2 ± 76.28 in Hela A1 and K1, respectively. It could be speculated that these cells have reduced their nuclear size accordingly to maintain the nuclear cytoplasmic (N/C) ratio. However, whether this effect may reflect some specific role of GRK2 in the processes of nuclear size control is highly speculative and remains to be established.

  • In line 418, is "Hur" "HuR"?

Thanks for noting this error, which has been corrected in the now line 429 of the revised version

Reviewer 2 Report

1- I don't recommend using of abbreviations in the title of a paper especially if the manuscript is not written only for specialists. 2- The aim of the work and according to the title of the manuscript are making concentration in a definite way on hypoxia and adrenergic stress, in relation to GRK2, HuR and HIF1α in cancer cell. Therefore, I suggest the following: a- The introduction to be formulated in a way that emphasizes harmony and orchestration of the three themes; GRK2, HuR and HIF1α by mentioning the role of each one alone in cancer cell and then making a tie or correlation between them in the presence of normoxic and hypoxic situations especially in tumors. b- I suggest to mention a separate paragraph for the adrenergic stress in relation to cancer and how a  possible correlation between adrenergic stress , cancer and the three themes pave the way to the current study. c- The data are too much in the section of result that may make the reader confused. Section 2.5 in the result (GRK2 fosters the HuR/HIF-1α axis and a pro-lymphangiogenic response in normoxic breast luminal) can be cancelled from results d- The part related to estrogen receptor-related breast cancer can be deleted as it is not related to the title of the manuscript    e- The paper in general has too much data which are valuable but need to be reduced to avoid confusion of the reader f- I recommend for the authors to conclude a clinical implication for the too much data which can be reflected in clinical practice either to molecular diagnosis in cancer or targeted drug treatment. Once more, please editor accept my apology for delay in reviewing this manuscript.  

Author Response

Reviewer 2

1- I don't recommend using of abbreviations in the title of a paper especially if the manuscript is not written only for specialists.

Using the term HuR for this RNA-binding protein is widely accepted across different biomedical fields. Non-abbreviated complete name corresponds to ELAV (embryonic lethal, abnormal vision, Drosophila)-like 1 (Hu antigen R), so is not possible to avoid the use of this well-established denomination taking into account the space limitations for titles  Akin to HuR, the term HIF-1α, corresponding to Hypoxia-Inducible Factor-1α, is also widely used in titles in the field GRK2 G protein coupled receptor kinase 2) might be less obvious for non-specialized biomedical researchers, although again is present in the titles of many publications Anyway, all these terms are explained in the abstract. Given the word count limitations in the Title, we consider that these abbreviations are very difficult to avoid. However, if the reviewer and editors judge it necessary, we are of course open to incorporate the entire names (HuR, HIF and GRK2), or alternatively only that for  GRK2.

2- The aim of the work and according to the title of the manuscript are making concentration in a definite way on hypoxia and adrenergic stress, in relation to GRK2, HuR and HIF1α in cancer cell. Therefore, I suggest the following:

a- The introduction to be formulated in a way that emphasizes harmony and orchestration of the three themes; GRK2, HuR and HIF1α by mentioning the role of each one alone in cancer cell and then making a tie or correlation between them in the presence of normoxic and hypoxic situations especially in tumors.

Following the suggestion of the reviewer, we have more clearly mentioned functional correlations between HuR and cancer in this section. This new text is in line 52.

b- I suggest to mention a separate paragraph for the adrenergic stress in relation to cancer and how a possible correlation between adrenergic stress, cancer and the three themes pave the way to the current study.

Following the suggestion of the reviewer, we have detailed the link of beta-adrenergic stimulation and cancer in the context of hypoxia-related processes. This new text is in lines 60-62

c- The data are too much in the section of result that may make the reader confused. Section 2.5 in the result (GRK2 fosters the HuR/HIF-1α axis and a pro-lymphangiogenic response in normoxic breast luminal) can be cancelled from results

Section 2.5 demonstrate the functional consequences of the GRK2/HuR/HIF1α axis in a relevant pathophysiological model. We think this is an important Result´s section that deserves to be maintained in the manuscript

d- The part related to estrogen receptor-related breast cancer can be deleted as it is not related to the title of the manuscript

Respectfully, we do not agree with this assessment. Both GRK2 and HuR are key players in the progression of estrogen receptor-related breast cancer and both proteins are particularly up-regulated in a relevant proportion of patients within these types of tumors. During tumor progression, cells are exposed to hypoxic conditions inasmuch as cells growth and tumor vessels are poorly functional. Moreover, adrenergic stress is a common feature observed in cancer patients. The GRK2/HuR interplay would endow estrogen-dependent breast cancer cells with an optimized response to different types of stress, thus fostering tumor progression. In sum, we think that this cancer type has relevant connections with the terms hypoxia and adrenergic stress in the manuscript title.      

e- The paper in general has too much data which are valuable but need to be reduced to avoid confusion of the reader

We hope that the issues related to this concern have been clarified in our responses to the points above

f- I recommend for the authors to conclude a clinical implication for the too much data which can be reflected in clinical practice either to molecular diagnosis in cancer or targeted drug treatment.

We appreciate the importance this reviewer gives to our results, and, following her/his suggestion, we now further discuss some clinical implications of the GRK2/HuR/HIF1α in cancer. This new text is located in lines 458-460 , 462 and 464-465.

Reviewer 3 Report

Comments relating to: 

GRK2-dependent HuR phosphorylation regulates 2 HIF1α activation under hypoxia or adrenergic stress.

C. Reglero et.al.

This manuscript provides new insight into the role of the G protein-coupled receptor kinase (GRK2) in modulating HIF1a responses to hypoxia, with a specific focus on signatures and effects observed in MCF7 breast luminal cancer cells and data from breast cancer patients.

In general, the manuscript is well structured, and the data presented is extensive and convincing. The authors clearly demonstrate a novel functional relationship between GRK2/HuR and HIF-1α. This observation has potential clinical relevance for a wide range of solid tumours. The observation that this mechanism may facilitate survival of malignant cells in estrogen-positive luminal breast cancer before tumours develop hypoxic cores is particularly interesting, as is the observation that chronic over-activation of the adrenergic system may potentiate tumour progression via this axis.

While overall the combined data and conclusions are convincing there are issues relating to the way the data is presented, and particularly the content of figure legends. In order to ease interpretation, all figure legends should be self-explanatory, enabling the reader to fully understand the data without cross reference to the main text. This is not the case for several of the figures as currently presented. In addition, there appear to be some errors in labelling or formatting within figures which also require attention. However, these are mainly formatting or data processing issues that should not require further experimental work to be performed.

Points to address:

Figure 1:

  • WT5, A1 and K1 should be defined in the legend, even though they are defined in the main body of the text.

  • With respect to images shown in Panels C & D, why is there no visible band for WT/GST-HuR in panel D as seen in Panel C?

Figure 2:

  • Given blots are shown in panel B why is an equivalent blot not shown in panel A?

Figure 3:

  • Legend says scale bars for both panels A and B = 10μm. This is a little surprising as the cells shown in panel B seem considerably larger than some of those shown in panel A. In K1 24h cells are particularly small and there is no scale bar. Two magnifications (40X and 63X) are quoted in methods, is this the issue?

Figure 4:

  • Panel A: legend says, cells were cultured under hypoxia and HIF levels were measured at indicated times. However, no times are shown in panel A.

  • It may seem obvious, but it would be good to state in legend or figure that GAPDH is being used a s a negative control.

  • There is inconsistency in the use of HIF1 and HIF1a, it may be better to use HIF1a

  • EV not defined

Figure 5:

Why are different loading controls being used for cell lysates?

Panel D: nuclear intensity trends shown in the plot show significant differences, however, this is not apparent in the representative fluorescent images, all of which look rather similar.

Specific use of nuclear intensity measurements may not be optimal, would it not be better to be considering nuclear/cytoplasmic ratios or nuclear to whole cells intensities?

Figure 6:

MCF7-luc not defined or indicated as control.

Why is GRK2 level higher in MCF7-luc than MCF7GRK2 at 48h ISO?

Legend for Panel C is very unclear: C) HIF1-α and DAPI nuclear marker levels were analyzed by immunofluorescence. The plot shows Ctrl v/s ISO but images show Ctrl v/s hypoxia? Also, the plot does not indicate if there are any significant differences in the values observed. In which case is the observation significant.  

While I think these points should be addressed and changes made as required, I do not think this detracts from the overall validity of the data or the conclusions made.

I would therefore support publication of the manuscript after appropriate revision.

Author Response

Reviewer 3

This manuscript provides new insight into the role of the G protein-coupled receptor kinase (GRK2) in modulating HIF1a responses to hypoxia, with a specific focus on signatures and effects observed in MCF7 breast luminal cancer cells and data from breast cancer patients.

In general, the manuscript is well structured, and the data presented is extensive and convincing. The authors clearly demonstrate a novel functional relationship between GRK2/HuR and HIF-1α. This observation has potential clinical relevance for a wide range of solid tumours. The observation that this mechanism may facilitate survival of malignant cells in estrogen-positive luminal breast cancer before tumours develop hypoxic cores is particularly interesting, as is the observation that chronic over-activation of the adrenergic system may potentiate tumour progression via this axis.

While overall the combined data and conclusions are convincing there are issues relating to the way the data is presented, and particularly the content of figure legends. In order to ease interpretation, all figure legends should be self-explanatory, enabling the reader to fully understand the data without cross reference to the main text. This is not the case for several of the figures as currently presented. In addition, there appear to be some errors in labelling or formatting within figures which also require attention. However, these are mainly formatting or data processing issues that should not require further experimental work to be performed.

We appreciate the overall positive comments of the reviewer and his/her suggestions to improve this work.

Points to address:

Figure 1:

  • WT5, A1 and K1 should be defined in the legend, even though they are defined in the main body of the text.

As indicated, the definition of WT5, A1 and K1 has been incorporated in the legend of Figure 1 in lines 101-105

  • With respect to images shown in Panels C & D, why is there no visible band for WT/GST-HuR in panel D as seen in Panel C?

We agree that the intensity of the GRK2-phosphorylated band corresponding to GST-HuR in panels C and D is not equal, although the GST-HuR band in panel D is in our view clearly visible, displaying higher signal compared to the phospho-defective GST-HuR mutants in the adjacent lanes. The lower visibility of the WT/GST-HuR band in panel D is explained by the low-exposure of the autoradiography used for the picture, and the different batches of purified GST-HuR proteins, which in panel D caused a slightly smeared bands compared to panel C. We have re-scanned a high-exposed autoradiographic film to improve the signal of phosphorylated GST-HuR bands in the new panel D.  In addition, for the consideration of the reviewer we include an additional example of the phosphorylation of GST-HuR proteins by GRK2, supporting that these results were reproducible.

Figure 2:

  • Given blots are shown in panel B why is an equivalent blot not shown in panel A?

This panel involved a complex experimental setting in which 5 different cell lines (Hela, A1, k1, WT5 and Sh-GRK2) were simultaneously analyzed in a hypoxia (3 time points) plus a normoxic condition in duplicate. A total of 25 samples were immunoblotted each time grouped in 3-4 different gels. To get representative blots of all conditions in the same experiment was not straightforward. We have included in the new panel A representative blots showing the time course pattern of HuR levels upon hypoxia corresponding to each cell line separately.        

Figure 3:

  • Legend says scale bars for both panels A and B = 10μm. This is a little surprising as the cells shown in panel B seem considerably larger than some of those shown in panel A. In K1 24h cells are particularly small and there is no scale bar. Two magnifications (40X and 63X) are quoted in methods, is this the issue?

Thanks for noting this erro., in fact panel B has a different magnification. The bar scale in the images of panel B is 7μm vs 10μm in panel A. We have amended the description of bars in the legend of figure 3B in line 184.

Regarding the image of K1 24h in panel A, the scale bar has been incorporated in the figure, and this scale is 10μm as the others in panel A. As the reviewer has noted, Hela cells expressing catalytically inactive GRK2 show a different nuclear size. Indeed, different stable clones and pooled Hela cells overexpressing GRK2-S670A or GRK2-K220R mutants have smaller nuclei suggesting that is not an artefact of the selection processes, but a specific effect event linked to the scaffold functions of catalytically inactive GRK2.  We have quantified the nuclear area and found that nuclei of Hela A1 and K1 are 74.2 µm2± 16.2 (n= 41) and 74.6 µm2 ± 30.5 (n=54) versus parental Hela cells 166µm2± 40.1 (n= 54). Interestingly, we have previously reported that the whole cell area of Hela A1 and K1 was also lower. These cell lines displayed a defective cell spreading compared to Hela cells caused by hyper-acetylation of microtubules, as GRK2 (but not the A or K mutants) phosphorylates and activates the tubulin deacetylase HDAC6 (Lafarga V, Aymerich I, Tapia O, Mayor F Jr, Penela P. A novel GRK2/HDAC6 interaction modulates cell spreading and motility. EMBO J. 2012 Feb 15;31(4):856-69). Thus, cellular area was 548.58µm2 ±66.47 in Hela versus 380.93µm2 ± 31.96 and 337.71 µm2 ± 76.28 in Hela A1 and K1, respectively. It could be speculated that these cells have reduced their nuclear size accordingly to maintain the nuclear cytoplasmic (N/C) ratio. However, whether this effect may reflect some specific role of GRK2 in the processes of nuclear size control is highly speculative and remains to be established.

Figure 4:

  • Panel A: legend says, cells were cultured under hypoxia and HIF levels were measured at indicated times. However, no times are shown in panel A

There is no time-course treatment since cells were only challenged with 2h hypoxia. We apologize for this mistake. The legend text of figure 4A in line 204 has been corrected accordingly.

  • It may seem obvious, but it would be good to state in legend or figure that GAPDH is being used a s a negative control.

Following the suggestion of the reviewer the indication of use of GAPDH as negative control in panel B has been included in the legend in line 211

  • There is inconsistency in the use of HIF1 and HIF1a, it may be better to use HIF1a

The term HIF1 has been replaced by HIF1α throughout the text

  • EV not defined

The term EV has been defined as Empty Vector in the figure legend in line 208 and 215-216

Figure 5:

  • Why are different loading controls being used for cell lysates?

We thank the reviewer for noting the inclusion of such different loading controls (actin and GAPDH blots in the same panel) in Figure 5 as well as in Fig. 6A, as it may be confusing.  Routinely, nitrocellulose membranes of these experiments were immunoblotted to detect GRK2 and two markers for protein loading (usually Actin or Tubulin and GAPDH) in a sequential way and after stripping if necessary, to monitor the comparable overall expression of proteins in the stable cell lines. In the former 6A panel, Actin blots were selected as control for GRK2 blots for the picture of MCF7-luc and MCF7-luc-shGRK2 cells because in these particular experiments they were developed at the time of GRK2 blots in the autoradiography, while GAPDH data were subsequently obtained after stripping from the Actin blot (as shown in the corresponding uncropped blots in the Supplementary material section). However, in most cases GAPDH blots are probed before Actin and together with GRK2. However, irrespective of the order in GAPDH or actin probing, the data used for internal normalization in Figs. 5 and 6A were those from GAPDH. Following the suggestion of the reviewer, we have now changed of the loading control protein marker in the representative blots, so they are standardized to GAPDH in figure 5A and also in figure 6A (following the suggestion of Reviewer 1 in the same sense). As indicated above, this change does not involve modifications in data quantification or plotted data inasmuch as densitometric values of GRK2 were always normalized by GAPDH in both panels 5A and 6A.

Moreover, we have previous evidence that actin, GAPDH and tubulin show similar efficacies for correcting protein loading among stable MCF7 cells either over-expressing or silencing GRK2 compared to parental cells (Nogués L, Reglero C, Rivas V, Salcedo A, Lafarga V, Neves M, Ramos P, Mendiola M, Berjón A, Stamatakis K, Zhou XZ, Lu KP, Hardisson D, Mayor F Jr, Penela P G Protein-coupled Receptor Kinase 2 (GRK2) Promotes Breast Tumorigenesis Through a HDAC6-Pin1 Axis. EBioMedicine. 2016 Nov;13:132-145). Thus, panel 5C includes a representative blot of similar GRK2-S67A and GRK2-S670D induction levels in MCF7cells using tubulin as control loading.   

  • Panel D: nuclear intensity trends shown in the plot show significant differences, however, this is not apparent in the representative fluorescent images, all of which look rather similar.

Levels of HIF1α are increased in tumor cells by default, but the extent, intensity, intracellular localization and distribution of HIF-1α protein is heterogeneous among different tumor cell types, and also within the population of a particular cell line. Although luminal MCF7 cells express lower levels of tHIF-1α compared to basal breast cell lines or other tumor types, the heterogeneity of its nuclear HIF-1α levels is evident. There is a range of nuclear staining intensities in which parental MCF7 cells spontaneously distribute in basal conditions, with an average value of 364.4 (mean gray value arbitrary units) and maximum 721.8 /minimum 125.3 values (from three experiments). Cells in each experiment can be stacked in groups according to their intensities (0-100; 101-200; 200-300; etc) across the entire range, and then allocated into three categories: the stack of the average value, the stacks over the average and the stacks below the average value. A representative fluorescent image reflecting this varying heterogeneity in a single microscopy field is not straightforward. However, a detailed analysis of the field of parental cells in figure 5D shows cells with intensities around the average HIF-1α intensity, over the average and below. These examples are now clearly marked in the image, zoomed and indicated as reference for this experiment. The same approach was followed with the mean gray value arbitrary units recorded in MCF7-GRK2 and MCF7 sh-GRK2 cells. Although the average intensity of HIF1α was not significantly modified, the range of HIF1α levels and cell distribution did change (MCF7-GRK2 382.6, max 959.7 /min 105.9; MCF7 sh-GRK2 422.2, max 560.1 /min 290.9). In the presence of extra GRK2, more cells move toward the maximum, while in the absence of GRK2 more cells concentrate close to the average value. This change at the cell population level is difficult to recapitulate on a field of few cells.

  • Specific use of nuclear intensity measurements may not be optimal, would it not be better to be considering nuclear/cytoplasmic ratios or nuclear to whole cells intensities?

HIF-1α staining is primarily nuclear, although some cytoplasmic staining of HIF-1α can also be detected in some conditions, particularly in tumor cells as consequence of diverse signals that may regulate HIF-1α stability in normoxia. As mentioned before, in MCF7 cells most of the HIF-1α  protein is concentrated in the nucleus as consequence of either reduced degradation or enhanced synthesis in the cytoplasm coupled to a constitutive, oxygen- independent nuclear translocation. In contrast to the nuclear signal, changes in the cytosolic immunofluorescence of HIF-1α were no obvious between MCF7-GRK2 or MCF7-shGRK2 cells and parental cells either in basal conditions or upon ISO challenge. Thus, we considered only the nuclear intensity measurements as a surrogate of the transcriptional functionality of HIF-1α. All confocal images from simultaneously processed parental, MCF7-GRK2 and MCF7-sh-GRK2 cells in each immunofluorescence experiment were acquired under the same parameters (pinhole, laser gain, etc). Parameters were set according to the brightest sample and the same threshold limits were maintained for all images. We therefore believe that our nuclear measurements are reliable, and that their possible normalization by a much lower, unchanged cytosolic signal will not improve the analysis of these results.  

Figure 6:

  • MCF7-luc not defined or indicated as control.

The term MCF7-luc refers to the parental MCF7 cell line used for generation of stable cells (overexpressing GRK2 or silencing sh-GRK2 constructs). Thus, MCF7-luc represents control MCF7 cells and it has been mentioned accordingly in the figure legend in line 322

  • Why is GRK2 level higher in MCF7-luc than MCF7GRK2 at 48h ISO?

Levels of GRK2 in MCF7-GRK2 are higher than those in MCF7-Luc (control parental cells) both in basal and ISO-stimulated conditions. Both MCF7-Luc (parental) and MCF7-GRK2 cells respond to 48h ISO upregulating GRK2 protein levels in a similar extent (fold) compared to their own basal levels, although the overall kinase levels remain higher in MCF7-GRK2 cells after treatment. In order to better show and quantify such fold increase, less exposed GRK2 blots were selected for the panel of MCF7-GRK2 cells compared to those in the panel of parental MCF7-luc cells. We acknowledge that there is not enough auto-explanatory information in these panels, what may lead to the question raised by the reviewer. Thus, we have completed the blot panel of MCF7-GRK2 with the condition of basal parental MCF7-luc cells (already present in the uncropped membrane) to provide an internal reference of endogenous unstimulated GRK2 levels, and showed the developed GRK2 signal both at higher and lower exposures in the new Figure 6A.

  • Legend for Panel C is very unclear: C) HIF1-α and DAPI nuclear marker levels were analyzed by immunofluorescence. The plot shows Ctrl v/s ISO but images show Ctrl v/s hypoxia? Also, the plot does not indicate if there are any significant differences in the values observed. In which case is the observation significant.

Thanks for noting this mistake. In the panel of immunofluorescence images (figure 6C) “normoxia” must read “Control” and “hypoxia” must read “ISO”.  These errors have been corrected accordingly. Statistical analysis of graph 6C has been performed as suggested by the reviewer.  We find a significant difference in HIF-1α levels between control and ISO-treated MCF7-GRK2 cells (p=0.0327, Student’s t-test). This significance has been indicated in the figure 6C.  

While I think these points should be addressed and changes made as required, I do not think this detracts from the overall validity of the data or the conclusions made.

I would therefore support publication of the manuscript after appropriate revision.

Round 2

Reviewer 1 Report

The revision has been done very well. But I have a comment.

<Minor comment>

In the response letter, the authors explained how the localization of HuR and HA signals were analyzed, and showed high magnification images of HuR and HA staining in which nuclei were indicated with dotted circles. These images should be added, for example in the supplementary materials. In addition, please describe the quantification procedure in the Methods section.

Author Response

Comments and Suggestions for Authors

The revision has been done very well. But I have a comment.

We appreciate the positive evaluation of the reviewer

Minor comment

In the response letter, the authors explained how the localization of HuR and HA signals were analyzed, and showed high magnification images of HuR and HA staining in which nuclei were indicated with dotted circles. These images should be added, for example in the supplementary materials. In addition, please describe the quantification procedure in the Methods section.

Following the indications of the reviewer, details of the quantification procedure of HuR and HA staining have been included in the Methods section (lines 508-515). Representative images showing the quantification method are incorporated in the supplementary material as a new Supplementary figure S4.